An evolving computational platform for biological mass spectrometry: workflows, statistics and data mining with MASSyPup64

Winkler Robert robert.winkler@ira.cinvestav.mx
Department of Biotechnology and Biochemistry, CINVESTAV Unidad Irapuato , Mexico
Ettrich Rudiger
Electronic publication date: 2015 Nov 17
Publication date: 2015
Volume: 3
Electronic Location ID: e1401
Received 2015 Sep 7; Accepted 2015 Oct 22
Copyright: © 2015 Winkler
Copyright year: 2015
Copyright holder: Winkler
License: This is an open access article distributed under the terms of the Creative Commons Attribution License, which permits unrestricted use, distribution, reproduction and adaptation in any medium and for any purpose provided that it is properly attributed. For attribution, the original author(s), title, publication source (PeerJ) and either DOI or URL of the article must be cited.
License URL: https://creativecommons.org/licenses/by/4.0/

Keywords: Bioinformatics, Computational mass spectrometry, Workflow decay, Metabolomics, Workflow management systems, Data Mining, Proteomics, Model building, Association analyses, Random forest trees

Funding: CONACYT basic science I0017/CB-2010-01/151596 FINNOVA I010/260/2014 CINVESTAV The study was funded by the CONACYT basic science grant I0017/CB-2010-01/151596, FINNOVA I010/260/2014 and the CINVESTAV. The funders had no role in study design, data collection and analysis, decision to publish, or preparation of the manuscript.

==============================
In biological mass spectrometry, crude instrumental data need to be converted into meaningful theoretical models. Several data processing and data evaluation steps are required to come to the final results. These operations are often difficult to reproduce, because of too specific computing platforms. This effect, known as ‘workflow decay’, can be diminished by using a standardized informatic infrastructure. Thus, we compiled an integrated platform, which contains ready-to-use tools and workflows for mass spectrometry data analysis. Apart from general unit operations, such as peak picking and identification of proteins and metabolites, we put a strong emphasis on the statistical validation of results and Data Mining. MASSyPup64 includes e.g., the OpenMS/TOPPAS framework, the Trans-Proteomic-Pipeline programs, the ProteoWizard tools, X!Tandem, Comet and SpiderMass. The statistical computing language R is installed with packages for MS data analyses, such as XCMS/metaXCMS and MetabR. The R package Rattle provides a user-friendly access to multiple Data Mining methods. Further, we added the non-conventional spreadsheet program teapot for editing large data sets and a command line tool for transposing large matrices. Individual programs, console commands and modules can be integrated using the Workflow Management System (WMS) taverna. We explain the useful combination of the tools by practical examples: (1) A workflow for protein identification and validation, with subsequent Association Analysis of peptides, (2) Cluster analysis and Data Mining in targeted Metabolomics, and (3) Raw data processing, Data Mining and identification of metabolites in untargeted Metabolomics. Association Analyses reveal relationships between variables across different sample sets. We present its application for finding co-occurring peptides, which can be used for target proteomics, the discovery of alternative biomarkers and protein–protein interactions. Data Mining derived models displayed a higher robustness and accuracy for classifying sample groups in targeted Metabolomics than cluster analyses. Random Forest models do not only provide predictive models, which can be deployed for new data sets, but also the variable importance. We demonstrate that the later is especially useful for tracking down significant signals and affected pathways in untargeted Metabolomics. Thus, Random Forest modeling supports the unbiased search for relevant biological features in Metabolomics. Our results clearly manifest the importance of Data Mining methods to disclose non-obvious information in biological mass spectrometry . The application of a Workflow Management System and the integration of all required programs and data in a consistent platform makes the presented data analyses strategies reproducible for non-expert users. The simple remastering process and the Open Source licenses of MASSyPup64 (http://www.bioprocess.org/massypup/) enable the continuous improvement of the system.

Introduction

Mass spectrometry provides qualitative and quantitative data about molecules. Since complex mixtures can be analyzed with high sensitivity and selectivity, mass spectrometry plays a central role in high-throughput biology (Jemal, 2000; Nilsson et al., 2010). Sequencing technologies have revolutionized the so-called ‘-omics’ sciences on the level of nucleic acids, genomics and transcriptomics (Sanger & Coulson, 1975; Wang, Gerstein & Snyder, 2009). But the study of the actual state of proteins and metabolites, which reflect the physiological condition of an organism, still relies mainly on mass spectrometry data.

In proteomics, a combination of biochemical and instrumental techniques is used to obtain comprehensive, quantitative information about the expression, modification and degradation of proteins at a certain physiological state (Wilkins et al., 1996; Anderson & Anderson, 1998). Although gel electrophoresis, immuno-precipitation and other separation strategies are used as first focusing steps, the identification of proteins usually relies on mass spectrometry methods (Shevchenko et al., 2006).

Metabolomics refers to the inventory of metabolites of an organism or tissue. The metabolome may be seen as an endpoint (Ernest et al., 2012), which derives from biochemical processes that depend on genomic and environmental factors. Therefore, the study of metabolic phenotypes allows both, the accurate classification of genotypes (Montero-Vargas et al., 2013; McClure, Chavarria & Espinoza, 2015; Musah et al., 2015) as well as an evaluation of the physiological state of an organism (García-Flores et al., 2012; García-Flores et al., 2015).

General mass spectrometry data processing workflow

The data analysis of mass spectrometry experiments all follow the same logic, although the composition of the samples, the analytical question and the data format and quality might vary. A general workflow in biological mass spectrometry is given in Fig. 1 and consists of the following steps:

Figure 1 Universal workflow of mass spectrometry data analyses.

Raw data need to be processed to extract informative features and to create statistically valid models.

Raw data import

First of all, the raw data need to be converted into a format which is readable for the following data analysis programs. This step is not trivial, since the different manufacturers of mass spectrometers use a variety of proprietary data formats. Currently, the recommended standard by the Human Proteome Organization (HUPO) Proteomics Standards Initiative working group for mass spectrometry standards (PSI-MS) is mzML (Martens et al., 2011). Therefore, most MS data analysis programs are able to read and process this format. The ProteoWizard tools (http://proteowizard.sourceforge.net) allow the conversion of vendor-specific files to mzML archives (Chambers et al., 2012; Kessner et al., 2008). Since format-specific libraries are required, it is recommend to execute the conversion to mzML files directly on the control computer of the mass spectrometer. Alternatively, the ProteoWizard software can be installed with the vendor-libraries on a Windows computer. The ProteoWizard tools without licensed and Windows-specific libraries are available on MASSyPup64 for further pre-processing of MS data files.

Spectra processing

Spectra are collected either in ‘profile’ mode or in ‘centroid’ mode. Profile spectra still contain the shape of peaks and thus may provide additional information about the measured compounds. However, the size of the data archives might be considerable, especially for high resolution measurements. In contrast, centroid spectra only consist of mass-to-charge (m/z) values and their intensity. In many cases, it is advised to convert profile spectra to centroid spectra, to reduce computing effort.

Typical operations of spectra processing include a baseline substraction, smoothing, normalization, and peak picking. On MASSyPup64, various programs are available for these tasks, such as: msconvert (Chambers et al., 2012), OpenMS/TOPPAS (Sturm et al., 2008) and R/MALDIquant (Gibb & Strimmer, 2012).

Some MS programs, such as Comet (Eng et al., 2015; Eng, Jahan & Hoopmann, 2013), X!Tandem (Craig & Beavis, 2004) and XCMS (Benton et al., 2008; Smith et al., 2006) do not require a prior external spectra processing, but can use raw mzML data as input.

Feature analyses

The mass spectrometry signals need to be transformed into chemical information. Therefore, ‘features’ have to be identified, which are e.g., defined by their m/z value and retention time. Usually the features display certain variations between samples, due to measurement tolerances. Those are corrected by an alignment of the feature maps, which finally allows to compare the abundance of features in different samples.

Different strategies permit the quantification of features: Label-free quantification, the evaluation of different ion transitions (fragments of a molecule in so-called Multiple-Reaction-Monitoring, MRM) or the use of defined tags.

The identification of molecules is desirable for most bioanalytical projects. For the identification of peptides and proteins, various search programs are available, which can be used or separately or in combination (Shteynberg et al., 2013). Identifying metabolites is still more challenging, although various databases, such as MassBank (http://www.massbank.jp/, (Horai et al., 2010)) and METLIN (https://metlin.scripps.edu/, (Smith et al., 2005)) as well as search algorithms have been published. The de-novo determination of chemical formulas from MS data is difficult, even with data from high-resolution instruments (Kind & Fiehn, 2006). Kind & Fiehn (2007) presented Seven Golden Rules (7GR) for the heuristic filtering of possible chemical formulas. The 7GR software was recently re-implemented for better usability and enriched with several functions. The respective program SpiderMass enables the construction of a custom data base with expected compounds for a certain biological context, which increases the probability of correctly identified metabolites (Winkler, 2015).

Statistics and Data Mining

Biological systems often exhibit notable variances, also measurement errors and wrong assignments of molecules are possible. Thus, usually biological and technical replicates are analyzed and the results are subjected to statistical analyses. More recently, Data Mining strategies are employed to unveil non-obvious information.

Different approaches for Statistics and Data Mining are presented below, as well as their practical application to proteomics and metabolomics data sets.

Integration and interpretation

In a last step, the information obtained has to be interpreted within a biological context. Changes of protein concentrations can indicate the involvement of physiological processes. Metabolic information can lead to information about pathways which are affected in certain conditions. Often, the identification of marker molecules is pursued, with the purpose to employ them later, e.g., for the early detection of diseases.

Statistics and Data Mining

The American Statistical Association describes Statistics as “the science of learning from data, and of measuring, controlling, and communicating uncertainty; and it thereby provides the navigation essential for controlling the course of scientific and societal advances” (http://www.amstat.org/, Davidian & Louis (2012)). Accepting this broad definition, Data Mining (DM) is a sub-discipline of Statistics.

Data Mining enhances ‘classic’ Statistics methods with machine learning (‘artificial intelligence’) algorithms and computer science. Data Mining supports the understanding of complex systems, which contain wealth of data with interacting variables. An important aspect of DM is the development of models, which represent the data in a structured form and support the extraction of information and creation of knowledge (Williams, 1987; Williams, 1988; Williams, 2011).

Creation of models can be distinguished into descriptive and predictive (Fig. 2).

Figure 2 Building of descriptive and predictive models.

Data can be transformed either into descriptive models, which integrate all available data, or into predictive models. For generating predictive models, only part of the data is used to train the model, another part to validate the model, and finally the model can be tested with the resting data to estimate error rates.

Descriptive models

Descriptive models analyze relationships between variables or between individual samples. Since these models search for structures in a given data set, they are developed using the whole data set. Two important strategies are:

• Association Analysis⋆

• Cluster Analysis⋆

Predictive models

Predictive models search for rules, which connect input and output variables. Those variables can be categorical (tissue type, color, disease/healthy) or numeric. If the target is categorical, the final model performs a Classification. If the target is numeric, a Regression. Important model builders are:

• Decision Tree Model⋆

• Random Forest Model⋆

• Support Vector Maschine (SVM) Model⋆

• Boost Model

• Linear Regression Model⋆

• Neuronal Network Model

For models marked with a ⋆, a practical example in proteomics and/or metabolomics is given below. For more details about the knowledge representation of DM models, their algorithms and examples we refer to Williams (2011).

Data Mining process and model development

Data Mining (DM) is mostly used in Economics, e.g., for managing risks of bank loans or for detecting fraudulent activities. However, the activities for developing a model is similar for any DM project. The Cross Industry Standard Process for Data Mining (CRISP-DM) defines six phases (Shearer, 2000):

1. Business Understanding

2. Data Understanding

3. Data Preparation

4. Modeling

5. Evaluation

6. Deployment

Obviously, in case of an Omics project we would replace ‘Business Understanding’ by ‘Problem Understanding’ or ‘Biological System Understanding’. The ‘Data Preparation’ is an important issue for analyzing mass spectrometry data. Depending on the number of samples and data quality, it might be necessary to eliminate variables or samples from the data set, to scale the data, to impute missing data points, etc. (Williams, 2011).

There is an important difference in the development of descriptive and predictive models. For descriptive models, the complete data set is used. For predictive models, the data set is separated into a training, a validation and a testing dataset, e.g., in a proportion 70:15:15 (Fig. 2). The training data serve for developing the model, the validation data set for monitoring the actual performance of the model, and the testing data for estimating the final performance of the model.

Final models can be exchanged between different computing environments using the XML based Predictive Model Markup Language (PMML) format (Grossman, Hornick & Meyer, 2002).

State of the art for statistics and DM in biological mass spectrometry

For proteomics, bioinformatic pipelines are already well established. The different peptide/ protein search engines deliver distinct scores, which indicate the confidence of a identification hit, such as the Mascot score, the e-value or the XCorr (Kapp et al., 2005; Becker & Bern, 2011). But independently of the employed MS/MS search program, a subsequent statistical anaysis is necessary. The PeptideProphet and ProteinProphet algorithms allow the statistical modeling of peptide and protein identification results (Nesvizhskii et al., 2003; Keller et al., 2002). Using target-decoy database searches permit the estimation of false discovery rates (Elias & Gygi, 2007). Commercial, as well as Open Source platforms, integrate those individual programs to create complete proteomic workflows (Nelson et al., 2011; Keller et al., 2005; Rauch et al., 2006; Deutsch et al., 2010; Deutsch et al., 2015). Finally, the submission of results in standard formats to public databases makes the data available to the community (Griss & Gerner, 2009; Barsnes et al., 2009; Vizcaíno, Foster & Martens, 2010; Côté et al., 2012; Vizcaíno et al., 2013; Mohammed et al., 2014; Reisinger et al., 2015; Killcoyne, Deutsch & Boyle, 2012; Desiere et al., 2006).

In metabolomics, still more issues are awaiting resolution. E.g., the unequivocal assignment of mass signals to the correct compounds and the estimation of the statistical confidence of metabolite identifications is still challenging.

The R packages XCMS/XCMS2 (Smith et al., 2006; Benton et al., 2008) and metaXCMS (Tautenhahn et al., 2011; Patti, Tautenhahn & Siuzdak, 2012) permit the realization of complete metabolic workflows and the comparison of various samples. Correct application of included functions improve the detection, quantification and identification of metabolites (METLIN database, (Benton, Want & Ebbels, 2010; Tautenhahn, Böttcher & Neumann, 2008; Smith et al., 2005)). The XCMS collection is technically mature and comprehensive, but for most casual users too complicated to handle. XCMS Online (Tautenhahn et al., 2012) facilitates the use of XCMS by non-experts. However, the control over data and the option to optimize the code for project-specific needs is limited in the online version.

MZmine 2 is another, java-based, framework for mass spectrometry data workflows with some statistical tools such as Principle Components Analysis (PCA) and Clustering capabilities, which is especially user-friendly and extensible (Pluskal et al., 2010).

Resuming, various bioinformatic solutions are already available for the processing and statistical analysis of proteomics and metabolomics data. But the concept of Data Mining is still not implemented in current biological mass spectrometry.

The traditional Omics approach is exploratory and starts from a biological question or problem. Usually it is rather curiosity- than hypothesis-driven. An Omics study normally ends with a statistically valid descriptive model, which is interpreted from a biological point of view. Often, the results help to build theories or hypotheses, which are testable afterwards.

In stark contrast, predictive models from Data Mining modeling can be immediately deployed and support decision making. Especially clinical applications (biomarker studies) and projects with limited sample availability (ecology, identification of microorganisms, ‘Biotyping’) could greatly benefit from the implementation of Data Mining strategies. Data Mining algorithms are also capable to uncover rules or patterns in complex data structures, without being biased by a (bio)scientist’s expectations.

Aim of this study

Data Mining strategies promise high potential for the analysis of biological mass spectrometry data, but there is still scarce use of it in current MS based Omics studies. On the other side, there is already a rich variety of excellent software for mass spectrometry data processing software (http://www.ms-utils.org/), and also for statistics and Data Mining available (Williams, 2011; Gibb & Strimmer, 2012; Luca Belmonte & Nicolini, 2013; Williams, 2009).

Thus, we compiled a computational platform for the high-throughput data analysis in proteomics and metabolomics, which facilitates the rapid set-up of workflows and the subsequent Data Mining. MASSyPup64 (http://www.bioprocess.org/massypup) is a 64-bit live system, which can be run directly from external media. Open Source licenses of the software and the remastering utility provided on Fatdog64 promote the further development and the adjustment to the needs of a laboratory.

Based on real datasets from proteomics and targeted and untargeted metabolomics we demonstrate the creation of efficient data processing workflows. Further, we stress out the opportunity to discover non-obvious biological knowledge by Data Mining methods in biological mass spectrometry.

Methods

Operating system

The original MASSyPup distribution was built on a 32-bit platform and contains multiple programs for analyzing mass spectrometry data Winkler (2014). The new MASSyPup64 is much more focused on the high-throughput processing of ‘big data’ and the subsequent Data Mining. MASSyPup64 is bootable on Windows (including with EFI ‘secure boot’) and Linux PCs.

As starting point, the 64-bit Linux distribution Fatdog64 was chosen (http://distro.ibiblio.org/fatdog/web/). The system is preferably installed on DVD or USB media. Data from all available local drives are accessible for analysis.

The mass spectrometry programs, special libraries and additional tools were compiled, if necessary and installed in the directory branch of /usr/local. Most programs can be started directly from a console window.

For Python (https://www.python.org/), versions 2 and 3 are installed. The default Python 2 is called by python, version 3 by python3.

Fatdog64 contains already a remastering tool with Graphical User Interface. Since the MASSyPup64 version already occupies several Gigabytes, it is recommendable to choose the “small initrd” option.

The current release of MASSyPup64, as well as FNAs (Frequently Needed Answers) and a list of currently installed software can be found on the project homepage (http://www.bioprocess.org/massypup/). All components are Open Source software, which permits the free distribution and modification of the system.

Workflow management systems

The ideal Workflow Management System (WMS) should be visual, modular and easy to understand. The facile integration of external commands and the development of new functions should be possible. Further, the WMS should allow to analyze data, which are stored outside the running platform, i.e., without uploading the data to the WMS. The last requirement is important, since mass spectrometry projects often exceed various Gigabytes of data volume and thus copying or moving them is inconvenient.

After trying various options, two WMS were installed on MASSyPup64:

1. TOPPAS

2. Taverna

The Trans-Proteomic-Pipeline 4.8.0 was compiled and installed on MASSyPup64, but without configuring the hosting server. Consequently, the TPP tools are available for being employed in workflows, but the web-interface is not running. Below, a workflow emulating the TPP for protein identification and validation is demonstrated.

Statistics and Data Mining

For statistical analyses, Data Mining and graphics, we compiled and installed an ‘R’ software environment (https://www.r-project.org/). A large scientific community is contributing to this powerful programming language (The R Journal, http://journal.r-project.org/).

Above mentioned XCMS/XCMS2/metaXCMS (Smith et al., 2006; Benton et al., 2008; Tautenhahn et al., 2011; Patti, Tautenhahn & Siuzdak, 2012) packages were installed, as well as MALDIquant/MALDIquantForeign for spectra processing (Gibb & Strimmer, 2012) and MSI.R for evaluating Mass Spectrometry Imaging (MSI) data (Gamboa-Becerra et al., 2015).

For the linear model analysis of metabolomic data, we included MetabR (http://metabr.r-forge.r-project.org/), which provides a Graphical User Interface (GUI) and can be used for both, statistical data evaluation and data preparation for Mining.

Rattle—the R Analytical Tool To Learn Easily—represents a sophisticated and free environment for Data Mining (http://rattle.togaware.com/, Williams, 2011; Williams, 2009). The GUI facilitates the loading, visualization and exploration of data, especially for beginners without profound R knowledge. Rattle also supports the export of models in PMML standard format and was installed on MASSyPup64 with all suggested packages (including database connectors, ggobi http://www.ggobi.org/, etc.).

Special tools for large data set editing and shaping

Standard spreadsheet software such as Excel, LibreOffice Calc or Gnumeric become very sluggish, if it comes to the editing of large tables. With R, huge tables can be handled and various of GNU programs (http://www.gnu.org/manual/blurbs.html), such as grep, sed, wc, .. can be used to edit big data files. But the import and manipulation of data is not always very practical with those tools. Therefore, some special programs for data manipulation were included into the MASSyPup64 distribution.

Spreadsheet program teapot

The non-traditional ‘Table Editor And Planner, Or: Teapot!’ was originally developed by Michael Haardt and Jörg Walter and is currently hosted at SYNTAX-K http://www.syntax-k.de/projekte/teapot/. For best performance and usability, Teapot was re-compiled and statically linked with the FLTK GUI toolkit (http://www.fltk.org).

Large matrix transposing

Frequently, it is necessary to transpose a data matrix before loading it into another program. This can be efficiently done with the command ‘transpose’ (version 2.0 by Dr. Alex Sheppard, http://www.das-computer.co.uk). The C program was modified and re-compiled to change the default maximum matrix size to 100,000 × 100,000.

Results and Discussion

Operating system and installed programs

Based on the Linux platform Fatdog64 (http://distro.ibiblio.org/fatdog/web/), an analysis framework and programming environment for mass spectrometry data was created. Table 1 lists the programs currently installed on MASSyPup64 (standard software such as editor, browser etc. are included as well, but not explicitly mentioned in the table). For installation, the iso image has to be burned on a DVD or installed onto a USB stick (e.g., with Rufus from https://rufus.akeo.ie). Upgrading individual programs or libraries is not necessary, because each release is a complete compilation of operating system and software. User settings, programs and data may be stored on local devices and are kept for future releases.

Table 1 Programs installed on MASSyPup64.

Development tools and libraries	
devx	Development tools for fatdog64-701, such as C/C++, FORTRAN compiler, header libraries	
jdk	Java Development Toolkit	
java	Version 1.7.0_65	
Java(TM) SE Runtime Environment	Build 1.7.0_65-b17	
Java HotSpot(TM) 64-Bit Server VM	Build 24.65-b04, mixed mode	
Python	Version 2 (python) and version 3 (python3)	
Workflow management	
taverna	Java-based workflow management with additional functions (condition, iteration)	
OpenMS/TOPPAS	Framework for mass spectrometry data processing and GUI supported design of pipelines	
Data conversion and trimming	
ProteoWizard tools	MS data conversion and tools, see /usr/local/massypup64/proteowizard-tools.txt; without vendor libraries	
transpose	Transposing matrix data from commandline, source code in /usr/local/bin/transpose.c. Currently compiled for a default maximum matrix size of 10,000 × 10,000	
teapot/fteapot	Spreadsheet program supporting three dimensional data sets. The manual “teapot.pdf” is located in /usr/local/doc	
MS data processing suite	
MZmine	Java-based MS analysis program, focused on metabolomics	
Proteomics search engines	
Comet	MS/MS protein search algorithm	
OMSSA	MS/MS protein search algorithm	
X!Tandem	MS/MS protein search algoritm	
TPP 4.8	Trans-Proteomic-Pipeline	
R statistics language	
XCMS/metaXCMS	Metabolomics	
Metab.R	Statistical analysis of metabolomics results	
Rattle	Data Mining	
Compound identification	
massXpert	Analysis of polymer spectra	
SpiderMass	Target DB creation/ matching, Online searches and formula generation	
Mass Spectrometry Imaging (MSI)	
HelloPhidget	Test tool for detection of connected Phidgets (prototyping of MSI)	
OpenMZxy	Control of Phidget imaging robot	
MSI.R	MSI analysis with R scripts	
Data visualization	
GGobi	Interactive data visualization and exploration tool	

If additional programs are required, they can be installed and included into a new MASSyPup64 version by using the Fatdog64 remastering tool.

Proteomics: identification of proteins, peptideprophet/proteinprophet validation, text mining and association analysis

Data set and bioanalytical question

Peroxidases are related to the post-harvest insect resistance of maize kernels (Winkler & García-Lara, 2010; García-Lara et al., 2007). Therefore, protein fractions of highly insect-resistance maize kernels with peroxidase activity were subjected to 1D or 2D gel electrophoresis and subsequently analyzed with LC-MS/MS. The data set consists of three samples: 2DM, a spot from 2D gel electrophoresis of maize kernels with peroxidase activity; 1DM, a protein band from 1D gel electrophoresis of partially purified peroxidase, and 1DR, a protein band with peroxidase activity from recombinant production of a putative peroxidase, which was cloned from cDNA. Details about the project can be found at López-Castillo et al. (2015).

The workflow should identify potential candidates of peroxidases and suggest peptides for a targeted screening of peroxidases.

Taverna Workflow Design

The design of the proteomics workflow using taverna was inspired by the work of Bruin, Deelder & Palmblad (2012), but several modifications were undertaken:

Peptide search

The search engine X!Tandem (Craig & Beavis, 2004) was replaced by Comet (Eng et al., 2015; Eng, Jahan & Hoopmann, 2013), in order to simplify the configuration by the user. All necessary parameters for the peptide identification are defined in the comet.params file, which has to be located in the same directory as the raw data files, which are expected in mzML format. A template for the comet.params file can be created by invoking the command comet -p. The location of the protein (or DNA) database is set with the database_name option. For performing a concatenated decoy search (Elias & Gygi, 2007),the parameter decoy_search needs to be set to 1. The separate generation of a decoy database is not required anymore.

PeptideProphet/ProteinProphet validation

The results of the Comet search are written directly to pep.xml format and can be passed to the PeptideProphet script (Keller et al., 2002). The processed pep.xml files are subsequently evaluated using ProteinProphet (Nesvizhskii et al., 2003). Both validation programs are part of the Trans-Proteomic-Pipeline (TPP) (Keller et al., 2005; Deutsch et al., 2010; Deutsch et al., 2015) and integrated into the workflow by very simple tool modules, which facilitate the modification of parameters by advanced users.

Creation of output in different formats

After the writing of the validated prot.xml files, the results are exported into various formats for further evaluation: The spreadsheet format xls (compatible with gnumeric and EXCEL), coma separated values (csv) text files, html (for opening the results in an internet browser) and in mzid (mzIdentML), a standard format for reporting proteomics results. The used tools were adopted from the TPP, the OpenMS/TOPPAS framework and from the Linux system programs, which underlines the flexibility of the taverna WMS.

Text extraction

In the last module, protein hits, which contain the defined search pattern for proteins of interest, are written into a separate summary file in csv format. This simple Text Mining step allows the rapid screening for relevant identification results.

An illustration of the complete workflow is given in Fig. 3.

Figure 3 Proteomics workflow with validation of hits by PeptideProphet/ ProteinProphet and final extraction of hits for proteins of potential interest.

For running the workflow (/usr/local/massypup64-taverna-workflows/m64-comet-val-export.t2flow), only the data path ‘/usr/local/massypup64-examples/ Maize-Proteomics-PODs’ needs to be given (as a value), and a string for the protein(s) of interest. In this example, “eroxidase” was defined as search string (omitting the initial letter P/p to avoid possible case problems).

A version of the workflow without the extraction module is also stored in the workflow examples directory. This workflow can be used for a batch-wise protein identification similar to the Trans-Proteomic-Pipeline. The short workflow only requires the mzML data/comet.params directory as input value.

Workflow results

Running the workflow delivers the expected output files, as well as sensitivity vs. error plots for the individual samples (see Fig. 4). Table 2 corresponds to the exported hits of putative peroxidades.

Figure 4 Plot of estimated sensitivity vs. error for sample 1DM, as delivered by the taverna workflow.

Figure modified from original program output for improved readability.

Table 2 Identified putative peroxidases, after PeptideProphet/ ProteinProphet validation.

Sample	Accession	Protein Prob.	Coverage	Unique peps	Description	
2DM	B4FBY8	0.6181	5.9	1	Peroxidase	
1DM	B4FK72	1.0000	2.7	2	Peroxidase	
1DM	B6T173	1.0000	12.7	7	Peroxidase	
1DM	K7TID5	1.0000	39.5	24	Peroxidase	
1DM	K7TID0	0.9937	9.5	1	Peroxidase	
1DM	B4FY83	0.9890	3.7	2	Peroxidase	
1DM	B4FNL8	0.0000		0	Peroxidase	
1DM	B6SI04	0.0000		0	Peroxidase	
1DM	K7VNV5	0.0000		0	Peroxidase	
1DR	K7TID5	1.0000	17.7	7	Peroxidase	
1DR	B6T173	0.9995	7.1	2	Peroxidase	
1DR	B4FSW5	0.9990	2.9	1	Peroxidase	
1DR	B4FY83	0.9990	3.7	1	Peroxidase	
1DR	K7TMB4	0.9990	3.3	1	Peroxidase	
1DR	Q6JAH6	0.6603	7.1	1	Glutathione peroxidase	
1DR	K7V8K5	0.5743	3.0	1	Peroxidase	
1DR	B4FNI0	0.3475	5.4	1	Peroxidase	
1DR	A0A0B4J371	0.0000		0	Peroxidase	
1DR	B4FBC8	0.0000		0	Peroxidase	
1DR	B4G0X5	0.0000		0	Peroxidase	
1DR	B6TWB1	0.0000		0	Peroxidase	
1DR	C0PKS1	0.0000		0	Peroxidase	
1DR	Q9ZTS6	0.0000		0	Peroxidase K (Fragment)	

Considering a minimum of two unique peptides and a probability of at least 0.95 as acceptance criteria, no peroxidase (POD) related protein was identified in the 2D spot, five POD candidates in the purified fraction from the 1D gel, and two PODs from the 1D gel after recombinant production of the putative POD B6T173 in Escherichia coli.

Thus, the workflow allows a rapid screening for proteins of interest. Indeed, further biochemical experiments confirmed protein B6T173 as the responsible one for the POD activity in the maize kernel.

Association analysis

Association Analysis investigates the probability of the co-occurrence of items. It is mainly known from Market Basket studies and social networks. For instance, if a person buys a telescope, most likely (s)he also might be interestd in a star map. Or if Henry knows Peter and Paul, he probably knows Mary as well. Importantly, the Association Analysis does not query the causality, but the likelihood of a relationship. Although the occurrence of an association might be low—let’s say the mentioned group of Henry, Peter, Paul and Mary represents a fraction of 0.0001% of a social network—the confidence might be high, e.g., 0.9, and therefore be highly informative.

To search out co-occurring peptides, which could lead to possibly associated proteins and suitable peptides for targeted proteomics, we carried out an Association Analysis with Rattle. In total, more than 700 peptides with a probability above 0.9 were considered. A minimal support of 0.6, a minimal confidence of 0.9 and a path of seven rules were chosen as parameters. Fig. 5 represents the associations between seven peptides, which are related.

Figure 5 Associated peptides across the samples.

Table 3 Association Analysis of peptides across three samples.

‘x’ stands for peptides present in the sample and ‘o’ for peptides missing in the sample.

Peptide	2DM	1DM	1DR	Acession	Description	
DSACSAGGLEYEVPSGRR	o	x	x	K7TID5	Peroxidase	
TDPSVDPAYAGHLK	o	x	x	B6T173	Peroxidase	
TVSCADVLAFAAR	o	x	x	B4FY83	Peroxidase	
VQVLTGDEGEIR	o	x	x	K7TID5	Peroxidase	
AFVHGDGDLFSR	x	x	x	B6SRJ2	Senescence-inducible chloroplast stay-green protein	
LFLNLQKEMNSVMVTRK	o	x	x	A0A096PYN5	30S ribosomal protein S2, chloroplastic	
GSGGGGGGGGGQGQSR	x	x	x	A0A096RDU5	Uncharacterized protein	

Table 3 lists the associated peptides together with their identifications. DSACSAGGLEYEVPSGRR, TDPSVDPAYAGHLK, VQVLTGDEGEIR are genuine peptides of B6T173. The peptide TVSCADVLAFAAR is not present in the amio acid sequence of B6T173, but the similar peptide TVSCADIVAFAAR. Since B6T173 was recombinately produced in E. coli (sample 1DR), the identification of this peptide indicates an unexpected phenomenon during the MS measurement. However, the respective transitions might be useful for the quantification of the protein.

The appearance of chloroplast protein might be feasible for the maize derived samples, but are unlikely to reflect reality in the bacterial preparation of B6T173. The glycine-rich peptide can be found in many organisms and therefore does not contribute information.

Resuming, a set of 3+1 peptides was found, which are highly indicative for the protein B6T173. Since the protein is related to post-harvest insect resistance, those peptide MS transitions could serve for the screening of seeds. The PeptidePicker workflow delivers theoretical peptides for targeted proteomics (Mohammed et al., 2014). However, if experimental data are available, the Association Analysis includes all possible variables which affect the peptide/ protein identification from sample extraction to final evaluation, and thus should suggest more reliable candidates.

Ideally, an Association Analysis is carried out with numerous individual samples, which allows to reduce the support limit and to bring to light non-obvious correlations between variables or observations.

Apart from finding reliable peptides for protein quantification, Association Analyses can be employed to discover alternative biomarkers, e.g., if the genuine biomarker is difficult to detect, or to search for protein–protein interactions.

Targeted metabolomics: cluster analyses, linear model analysis and model building using data mining

Data set and analytical question

We re-processed a data set which was described by Ernest et al. (2012). To study the adipose tissue metabolism, three groups of chicken where analyzed, which underwent different treatments: A control group (“Control”, sample 1–7) which were fed ad libitum, chicken fasted for 5 h (“Fast”, sample 8–14) and a group treated with an insulin inhibitor (“InsNeut”, sample 15–21). For more details about the biological experiments, we refer to the original paper Ernest et al. (2012). From the targeted metabolomics data, a statistical analysis yielding fold-changes and p-values should be carried out. Further, a classification of the three groups, based to their metabolic profile, should be intended.

Statistical evaluation with MetabR

Using MetabR, the fold-differences and the Tukey’s Honest Significant Difference (HSD) was calculated, applying a fixed linear model for the variables “Quantity” and “Internal Standard” (Table 4). The script also performs an Hierarchical Cluster Analysis (HCA, Fig. 6).

Table 4 Statistical analysis of targeted metabolomics data with MetabR.

Bold values are significant with p-values <0.01 (Tukey HSD).

	Fast-control	InsNeut-control	InsNeut-fast	
	Fold	p-value	Fold	p-value	Fold	p-value	
ATP	1.27	0.38	1.06	0.93	0.83	0.59	
Citraconate	1.08	0.25	1.05	0.56	0.97	0.81	
Citrate	1.22	0.08	1.00	0.96	0.82	0.13	
Dihexose	0.08	<0.01	0.59	0.93	7.22	<0.01	
Inosine	0.74	0.33	0.91	0.58	1.24	0.89	
Lactate	0.87	0.14	0.99	0.97	1.14	0.20	
Pyruvate	1.20	0.19	0.97	0.95	0.81	0.11	
2-Oxoglutarate	0.93	0.75	1.51	<0.01	1.63	<0.01	
1-Methyladenosine	1.20	0.99	1.13	0.96	0.95	0.99	
Glutamine	0.68	0.03	2.51	<0.01	3.71	<0.01	
Guanosine	0.76	0.22	0.83	0.26	1.09	0.99	
O-Acetyl-L-serine	0.59	0.30	2.13	0.11	3.62	<0.01	
Glucosamine	1.36	0.22	2.98	<0.01	2.20	<0.01	
Thiamine	0.54	0.14	0.89	1.00	1.66	0.14	

Figure 6 Hierarchical Cluster Analysis (HCA) of targeted metabolomics from chicken groups.

Figure modified from original program output for improved readability.

For Dihexose, 2-Oxoglutarate, Glutamine, O-Acetyl-L-serine and Glucosamine significant differences of the metabolite concentrations were stated. In the Hierarchical Cluster Analysis (HCA), the fasting chicken and the chicken treated with insulin inhibitor are separated (Fig. 6). The control chicken samples are found in both branches of the dendrogram, which indicates that (a) the clustering method is not selective enough to clearly separate the samples based on their metabolic identity or (b) that the metabolic profiles of the control group is to divers to be classified correctly.

The results of the statistical analyses are in agreement with the original publication by Ernest et al. (2012). However, to improve the classification of the three groups we probed alternative approaches for Clustering and Model Building.

Clustering approaches and their limitatations

Clustering helps to identify similar groups in a data set. Estimating the adequate number of clusters is not trivial and various algorithms have been described for this task. We tested several of them, which are available within R (http://stackoverflow.com/questions/15376075/cluster-analysis-in-r-determine-the-optimal-number-of-clusters/15376462#15376462). The different plots can be reproduced with the cluster-chicken.R script located in the /usr/local/massypup64-examples/Chicken-Data-Mining directory.

K-means clustering and sum of squared error (sse) plot

The K-Means Clustering method of Hartigan & Wong (1979) is implemented in the R function kmeans and minimizes the sum of squared errors between data points. Since three clusters are expected from the biological context, we performed a K-Means cluster analysis with a starting value of ‘3’. As shown in Fig. 7A, no clear separation of the three chicken groups was achieved.

Figure 7 (A) K-Means clustering of the normalized chicken data set, considering three clusters, (B) SSE plot for estimating the cluster number.

The corresponding SSE plot is lacking a local minimum (‘elbow’), which would indicate the optimum number of clusters in the data set (Fig. 7B). The SSE plot indicates that K-Means Clustering based on the minimization of the Sum-of-Square-Error is not suitable for classifying the three chicken groups.

Silhouette plot and silhouette plot based clusters

Silhouettes help in the graphical evaluation of clustering solutions and in the choice of an adequate number of clusters (Rousseeuw, 1987). The resulting graphs in Fig. 8 are also based on a K-Means Clustering and suggest two clusters.

Figure 8 (A) Silhouette plot and (B) Silhouette plot based clusters.

Caliński–Harabasz Index

The Caliński–Harabasz Index (Caliński & Harabasz, 1974) demonstrated excellent recovery and consistent performance in a comparative study of Milligan & Cooper (1985). Therefore, the cascadeKM function of the R package vegan was used for a Caliński–Harabasz analysis.

Figure 9 Estimation of the number of clusters using the Caliński-Harabasz index.

The resulting graphs (Fig. 9) indicate indeed a minimum for three clusters. But the number of objects in each group is not congruent with the individual chicken in each group.

Affinity Propagation (AP) Clustering

Frey & Dueck (2007) proposed the Affinity Propagation (AP) Clustering algorithm, in which information is exchanged between data points until an optimal solution is reached. The algorithm is computationally efficient and more accurate compared to other strategies. We applied the R function apcluster to the data matrix and the transformed data matrix.

Figure 10 (A) Affinity propagation (AP) clustering and (B) AP clustering with transformed data matrix.

AP clustering yields four clusters for the chicken groups (Fig. 10A). The insulin inhibitor treated chicken (objects 15–21) cluster together. However, there is also another sample from the control group in the same branch. The clustering of the transformed data matrix suggests correlations between three groups of metabolites (Fig. 10B), which could lead to related metabolic pathways.

MClust Algorithm

The R package mclust tries different probability models and plots the number of cluster elements versus the Bayesian Information Criterion (BIC) (Fraley & Raftery, 2002).

Figure 11 MClust analysis for testing different probability models.

The model labeled as ‘EVV’, which stands for ‘multivariate mixture model with ellipsoidal, equal volume’ displays the highest BIC values (Fig. 11). However, no maximum is reached for three cluster groups, which indicates that no tested model is suitable for a correct clustering.

Summary of clustering approaches

Table 5 summarizes the number of clusters, which was estimated by different algorithms. The Caliński–Harabasz index guesses the correct number of groups in the dataset, but no evaluated clustering method is specific enough to accurately separate the three chicken groups. Therefore, we continued with a Data Mining based model building.

Table 5 Comparison of methods for estimating the number of clusters in the targeted metabolomics dataset of three chicken groups.

Method	No. of clusters	
K-Means/SSE	n.a.	
Silhouette Plot	2	
Caliński–Harabasz	3	
Affinity Propagation clustering	4	
MClust algorithm	n.a.	

Data Mining based model building

Conveniently, the normalized data from the statistical evaluation with MetabR can be loaded directly into Rattle for Data Mining. For the supervised building of models, we split the data in a ratio of 70:20:10 for Training, Validation and Testing. As target value, the experimental group of the chicken with the categorical values “Control”, “Fast” and “InsNeut” was set. Following, the results for different models are presented. The performance of the models in the three stages of development is summarized in Table 6.

The models and supporting data are included in the MASSyPup64 examples; The Rattle sessions are stored in files with the extension .rattle.

Decision Tree

Decision Tree models result in simple representations, which are easy to understand and easy to put into practice. The Decision Tree model for classification was built using the R package rpart with 14 samples and yielded the following rule set:

n= 14 node), split, n, loss, yval, (yprob) * denotes terminal node 1) root 14 9 Control (0.3571429 0.3571429 0.2857143) 2) Dihexose >= 9.851921 9 4 Control (0.5555556 0.0000000 0.4444444) 4) X2.Oxoglutarate < 14.84659 5 0 Control (1.0000000 0.0000000 0.0000000) * 5) X2.Oxoglutarate >= 14.84659 4 0 InsNeut (0.0000000 0.0000000 1.0000000) * 3) Dihexose< 9.851921 5 0 Fast (0.0000000 1.0000000 0.0000000) *

Those rules can be used in their plain form or implemented into a simple computer program. The graphical representation is given in Fig. 12.

Figure 12 Decision tree model for the classification of chicken samples.

Both, the equation form and the graphical Decision Tree models are straight-forward to understand and deploy, e.g., for diagnostics applications. The evaluation of the model using an Error Matrix (Table 6) returns one error (33%) for the validation and one error (25%) for the testing data. All samples of the training set were identified correctly, resulting in an overall error rate of 9.5%.

For certain uses, such as models supporting medical decisions, a very low false-positive or false-negative rate is needed. If needed, the model can be optimized towards a certain direction, such as avoiding false-negatives (for details see Williams (2011)). Another option is the building of more complex models like Random Forest Tree or Support Vector Machine models.

Random forest tree model

For building a Random Forest Tree model, multiple Decision Trees are created and combined into a single model (Williams, 1988; Williams, 1987). Random Forest Models are characterized by robustness to noise, outliers and overfitting (Williams, 2011). An important aspect is also the selection of variables: only a part of the available variables—by default the square root of all variables—is used for each individual Decision Tree. In this ‘bagging’ strategy the same variable may occur more than once.

For building the Random Forest Tree, we defined the construction of 5,000 trees and three variables for each split. The ‘out-of-bag’ (OOB) error estimate is based on the observations, which are not considered in the training set and was calculated as 14.29%.

Strikingly, the Random Forest Tree Model classified all samples without error in any stage of development (Table 6).

Another result of the model building is highly informative: The Variable Importance (Fig. 13).

Figure 13 Variable importance from the Random Forest Tree modeling for the classification of chicken samples.

Figure 13A refers to influence of the variables on the predictive accuracy of the Decision Tree, the right plot expresses the impact on the Gini index (a measure of statistical dispersion (Gini, 1912)) when splitting on a variable. The first eight variables are equal in both measures, and indicate a high importance of the concentrations of Dihexose, Glutamine, X2.Oxoglutarate and Glucosamine. Those metabolites also show significant changes in the statistical analysis with MetabR (Table 4), but the Random Forest Tree analysis now allows for the correct classification of the samples and suggests an order for the importance of variables.

Support Vector Machine (SVM) and Linear Model

Several more model builders are available in Rattle, such as Neuronal Networks and the Boost algorithm. Because of their popularity in the community, we also tested the Support Vector Machine (SVM) and the Linear Model for the chicken dataset. The results are collected in Table 6.

The SVM model performed equally well as the Random Forest Tree model, i.e., in no stage of the development a sample was classified wrongly. In contrast, the Linear Model presented one error during the validation and one error in the testing.

Table 6 Error Matrix for predictive models, which were developed for the classification of chicken groups, based on targeted metabolomics data.

	TRAINING	VALIDATION	TESTING		
Decision Tree										Error	
	Predicted								
Actual	Control	Fast	InsNeut	Control	Fast	InsNeut	Control	Fast	InsNeut	0.25	
Control	5	0	0	0	0	1	1	0	0	0.0	
Fast	0	5	0	0	0	0	1	1	0	0.5	
InsNeut	0	0	4	0	0	2	0	0	1	0.0	
Random Forest											
	Predicted								
Actual	Control	Fast	InsNeut	Control	Fast	InsNeut	Control	Fast	InsNeut	0.0	
Control	5	0	0	1	0	0	1	0	0	0.0	
Fast	0	5	0	0	0	0	0	2	0	0.0	
InsNeut	0	0	4	0	0	2	0	0	1	0.0	
Support Vector Machine											
	Predicted								
Actual	Control	Fast	InsNeut	Control	Fast	InsNeut	Control	Fast	InsNeut	0.0	
Control	5	0	0	1	0	0	1	0	0	0.0	
Fast	0	5	0	0	0	0	0	2	0	0.0	
InsNeut	0	0	4	0	0	2	0	0	1	0.0	
Linear Model											
	Predicted								
Actual	Control	Fast	InsNeut	Control	Fast	InsNeut	Control	Fast	InsNeut	0.25	
Control	5	0	0	1	0	0	0	0	1	1.0	
Fast	0	5	0	0	0	0	0	2	0	0.0	
InsNeut	0	0	4	1	0	1	0	0	1	0.0	

Comparison of Model Builders and Cluster Analyses

The Support Vector Machine (SVM) and the Random Forest Tree strategy generated error-free models for the classification of the three chicken groups. This classification was not possible with Cluster Analyses, which suggests the use of Data Mining models for data sets with only subtle differences between experimental groups.

The Random Forest Tree model additionally delivers quantitative measures for the variable importance, which facilitates the discovery of biologically relevant factors.

Untargeted metabolomics: discovery of important variables by Data Mining and identification of putative metabolites

Data set and bioanalytical question

The data analysis for untargeted metabolomics experiments is highly complicated, since unknown metabolic features need to be detected and aligned between samples. To gain biological knowledge, these features need to be identified and integrated into metabolic pathways. Recently we reported the metabolic fingerprinting of the Arabidopsis thaliana accessions (‘ecotypes’) Columbia (“Col-0”) and Wassilewskija (“Ws-3”), based on extracts of leaves and inflorescence (Sotelo-Silveira et al., 2015). In this example, we re-process the reduced datasets of the inflorescence samples and try to identify possibly distinct pathways between the inflorescence samples of the two accessions.

Date pre-processing and TOPPAS pipeline for feature detection and alignment

The original mzML data were processed with msconvert to reduce noise signals and to reduce the size of the data files. Figure 14 represents the workflow for the data processing, which was implemented in TOPPAS. First, the MS features are detected in all data files. Following, the features of all samples are aligned and the results exported into a text file for further statistical analyses. The complete pipeline and (.toppas) the mzML raw data files are available in the example directory.

Figure 14 TOPPAS pipeline for MS feature detection and alignment, with output of the consensus features in a text file.

Statistical analyses and building of a random forest tree model

The 1,005 high-quality features, which were detected in all twelve samples, were normalized with MetabR and loaded into Rattle (as described before in the targeted metabolomics example). A Random Forest Tree model was built for the classifications of the accessions with default parameters, calculating 5,000 trees. The classification was correct (0% error rate) in all steps of the model development. This finding demonstrates again the high robustness and selectivity of the Random Forest Modeling for metabolomics data, which are usually characterized by many variables and few repetitions.

Important variables and identification of putative metabolites and pathways

The m/z values of features were matched with an Arabidopsis meta-database using SpiderMass (Winkler, 2015). Putatively identified metabolites were sorted by their Variable Importance (accuracy criterion), manually revised and assigned with their pathway or function (Table 7).

Table 7 Putative identifications for important variables for the classification of Arabidopsis, based on untargeted metabolomics profiles.

m/z	Variable importance	Ionization mode	Name	Function/pathway	Mass error [mDa]	
463.105	2.65	[M+H]+	7-Methylthioheptyl glucosinolate	Glucosinolate biosynthesis	4.6	
249.149	2.45	[M+H]+	Abscisic acid aldehyde	Abscisic acid biosynthesis	0.1	
249.149	2.45	[M+Na]+	Methyl Dihydrojasmonate	Aroma compound	2.5	
227.070	2.45	[M+Na]+	Tryptophan	Amino acid	−9.3	
202.090	2.00	[M+Na]+	L-Phenylalanine	Amino acid	5.8	
647.159	2.00	[M+Na]+	Isorhamnetin-3-O-rutinoside	Flavonoid glycoside	0.8	
245.099	2.00	[M+H]+	Biotin	Vitamin	4.0	
631.162	2.00	[M+Na]+	Diosmin	Flavonoid glycoside	−1.3	
387.025	2.00	[M+Na]+	Xanthosine 5’-phosphate	Purine metabolism	−6.0	
329.068	2.00	[M+Na]+	Leucocyanidin	Flavonoid	4.8	
221.031	2.00	[M+H]+	Imidazole acetol phosphate	Amino acid biosynthesis	−0.9	
633.141	1.73	[M+Na]+	Rutin	Flavonoid glycoside	−2.0	
223.169	1.73	[M+Na]+	Lauric acid	Fatty acid	2.4	
595.160	1.73	[M+H]+	Flavonoide glycoside (isobars)	Flavonoid glycoside	−5.4	
579.163	1.73	[M+H]+	Flavonoide glycoside (isobars)	Flavonoid glycoside	−7.7	
263.090	1.73	[M+H]+	2-(6’-Methylthio)hexylmalic acid	Glucosinolate biosynthesis	−6.2	
271.132	1.73	[M+Na]+	Abscisic acid aldehyde	Abscisic acid biosynthesis	1.3	
195.065	1.73	[M+H]+	Ferulic acid	Cell wall formation	−0.4	
251.021	1.73	[M+Na]+	Mevalonate 5-phosphate	Terpene biosynthesis	−7.9	
403.064	1.73	[M+Na]+	O-Acetylserine	Amino acid biosynthesis	−6.1	
331.158	1.73	[M+H]+	Gibberellin A5	Plant hormone	4.0	
457.044	1.73	[M+Na]+	5-Methylthiopentylglucosinolate	Glucosinolate biosynthesis	−7.1	
317.175	1.73	[M+H]+	Gibberellin A9	Plant hormone	0.1	
333.209	1.73	[M+H]+	Gibberellin A12	Plant hormone	2.6	
333.209	1.73	[M+Na]+	6,9-Octadecadienedioic acid	Fatty acid	5.0	
479.099	1.73	[M+H]+	Hyryl	Coenzyme (Riboflavin, FMN, FAD)	5.1	
479.099	1.73	[M+Na]+	Flavin mononucleotide (FMN)	Coenzyme	5.1	
625.174	1.41	[M+H]+	Narcisin	Flavonoid glycoside	−1.8	
245.042	1.41	[M+H]+	1,3,7-Trihydroxyxanthone	Xanthones	−2.7	
611.157	1.41	[M+H]+	Rutin	Flavonoid glycoside	−3.9	
601.147	1.41	[M+Na]+	Flavonoide glycoside (isobars)	Flavonoid glycoside	−5.5	
369.123	1.41	[M+Na]+	Gibberellin (isobars)	Plant hormone	−8.2	
349.058	1.41	[M+H]+	Inosinic acid	Ribonucleotid biosynthesis	3.6	
328.941	1.41	[M+Na]+	D-Ribulose 1,5-bisphosphate	Phothosynthesis	−4.9	
365.128	1.41	[M+Na]+	Abietin	Terpene	7.5	
369.124	1.41	[M+Na]+	Gibberellin (isobars)	Plant hormone	−7.3	
311.187	1.41	[M+H]+	Botrydial	Terpene	1.3	
385.014	1.41	[M+Na]+	Xanthosine 5’-monophosphate	Purine metabolism	−2.3	
433.118	1.41	[M+H]+	Apigenin glucoside	Flavonoid glycoside	4.7	
349.057	1.41	[M+H]+	Inosinic acid	Ribonucleotid biosynthesis	2.5	
221.042	1.41	[M+H]+	Imidazole acetol phosphate	Amino acid biosynthesis	9.6	
221.042	1.41	[M+H]+	2-(3’-Methylthio)propylmalic acid	Glucosinolate biosynthesis	−7.0	
221.042	1.41	[M+Na]+	Syringic Acid	Aminobenzoate degradation	−0.2	
625.170	1.41	[M+H]+	Narcisin	Flavonoid glycoside	−6.1	
349.200	1.41	[M+H]+	Gibberellin (isobars)	Plant hormone	−0.8	
363.039	1.41	[M+H]+	Xanthosine 5’-monophosphate	Purine metabolism	4.4	
211.057	1.41	[M+H]+	5-Hydroxyferulic acid	Phenylpropanoid biosynthesis	−3.0	

The results of affected metabolic pathways are congruent with the previously reported statistical analyses (Sotelo-Silveira et al., 2015). But taking into account the Variable Importance for the classification of the inflorescence profiles according to their accession, now allows a statistically supported ranking of putatively involved pathways. The biosynthesis of (thio)glucosinolate appears to be the most significant variable, followed by the biosynthesis of abscisic acid biosynthesis, an aroma compound, and amino acids. Most of the compounds down the list are related to plant hormones, flavonoid glycosides and cofactors.

Thus, the Data Mining method is not only a tool for the reliable classification of sample groups, but also supports the discovery and ordering of biologically relevant variables.

Conclusions

The presented examples from proteomics and metabolomics demonstrate the high potential of integrating Workflow Management Systems with Data Mining tools and helper programs into a single data analysis platform. The ready-to-use combination of software packages and the availability of data on the live system facilitates the repetition of the experiments and prevents workflow decay.

Data Mining strategies enhance the knowledge generation from biological mass spectrometry data. Predictive models can be readily deployed for future decision making, e.g., in clinical diagnostics. The Graphical User Interfaces (GUIs) of MetabR and Rattle enable the easy application of advanced Statistics and Data Mining for biological mass spectrometry data.

Association Analyses reveal relations between variables and can be used to search for interactions, which are present in low frequency, but with high confidentiality, e.g., in the search for co-occurring peptides or related proteins.

The Random Forest Tree models demonstrate high robustness and accuracy for the classification between experimental groups from metabolomics data. The variable importance supports the discovery and ranking of significant metabolites and pathways.

Data Mining paves the way for a deeper understanding of biological phenomena by a more profound analysis of mass spectrometry data. MASSyPup64 provides a stable and evolving platform for this challenging task.

Additional Information and Declarations

Competing Interests

Author Contributions

Data Availability

The author declares there are no competing interests.

Robert Winkler conceived and designed the experiments, performed the experiments, analyzed the data, contributed reagents/materials/analysis tools, wrote the paper, prepared figures and/or tables, reviewed drafts of the paper.

The following information was supplied regarding data availability:

http://www.lababi.bioprocess.org/index.php/lababi-software/78-massypup.

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
