# Peer review of "An evolving computational platform for biological mass spectrometry: workflows, statistics and data mining with MASSyPup64"

_PeerJ, doi:10.7717/peerj.1401_

## Round 0.1 · original submission · Major Revisions

Dear Robert,

Thank you for submitting your manuscript for publication in PeerJ. It has been examined by three expert reviewers who have concluded that the work is potentially sound and suitable for publication as the evolution of MassyPup64 is an important development and the manuscript offers an overview of the improved platform and presents specifically designed workflows that are useful to the community; however, it appears that all three reviewers consider the manuscript not yet clear enough especially for a reader that is unfamiliar with the software and still missing relevant information, which means that a revision will be needed prior to its further consideration for publication.

Please see the enclosed reviewer's reports for details regarding the requested changes and/or additions.

When preparing your revision, please focus mainly on the style and organization of the paper to improve clarity and the understanding what is included and what it can be used for. I consider the suggestion of reviewer 1 and reviewer 3 to include a full page table listing of the software included to be an idea that you should consider.

Please also take very serious the concern of reviewer one, mentioned in the "Validity of findings" about addressing the extremely high false-discovery rate.

Please address either in your revised manuscript the concerns/remarks raised by all three reviewers or in case you have objections to a specific recommendation please present your arguments in a point-to-point response.

·

Basic reporting

Review of: An Evolving Computational Platform for Biological Mass Spectrometry: Work-flows, Statistics and Data Mining with MASSyPup64

Getting a large number of packages to work together on a single operating system is a daunting task. Configuring specific compilers, interpreters, and organization of permissions is often not achievable by typical scientists. The evolution of MassyPup64 is an important development to encourage distribution of tools that can seemlessly work together.

Recommended revisions:
Original data is almost always in a proprietary format from a vendor. Elaborate on line 44 which formats are compatible with msconvert via the MassyPup64 Linux distribution and which are not. Its my naieve understanding that certain vendors have released their dlls in such a way that they are compatible with Wine and other Linux .net emulators. This is an excellent opportunity to identify the current state of affairs for raw data import for each vendor.

imzml/ibd files are mentioned in figure 1, but there is no mention of mass spectrometry imaging other than line 234 "MSI.R for evaluating MSI data." There are numerous software packages for processing imzml, and they aren't mentioned. This should be corrected unless there is a reason that they can't be installed on MassyPup64?

Python is mentioned briefly on line 205 and there is absolutely no mention of the IPython interface. Including the Python packages for mass spectrometry, computational chemistry, and scientific computing packages needed for plotting, statistics, and file import/export seems like a major oversight that could be easily overcome.

It isn't clear from reading the manuscript if an MSMS reference database (ie: from mass bank or Metlin) and a search algorithm for comparing a measured MSMS spectrum to a reference database is included.

It isn't clear if mzmine is included in this release. For example, mzmine is mentioned on line 162. Is it included? Is Java included? Is the java version compatible with other java apps in MassyPupp64?

My general impression is that many valuable tools are bundled together in this linux distribution, but the style and organization of the paper in its current form makes it very difficult to tell exactly what is included and what it can be used for. A full page table listing the software included would make this much clearer.

Experimental design

Not applicable to this manuscript

Validity of the findings

I was surprised to see the mention of putative-identification on line 503-505. I did not see any mention of the consolidation of features into pure-spectra where adducts, isotopes, and degradation products can be identified. It is a concern that this this manuscript advocates for accurate mass based assignments of features to compounds independent of pure-spectra and MSMS. This may not be the viewpoint of the author, but concluding the paper on this point makes it seem like a highlight. Addressing the extremely high false-discovery rate in this approach is necessary, at a minimum.

Reviewer 2 ·

Basic reporting

The manuscript by Robert Winkler details a computational platform for mass spectrometry analysis called MASSyPup64, which is an evolution on the already published MASSyPup platform. But while the work on providing easy access to open source bioinformatics software is highly commendable, the manuscript in its current form leaves me rather confused.

To me it seems like the author is trying to write multiple manuscripts in one, and I think that readers not familiar with most of the software used will be unable to extract much from the text. I'd recommend restructuring and refocusing the manuscript into more of a technical brief kind of manuscript, with a lot of the current details and examples moved to either a supplementary or the MASSyPup64 web page. I think this will make it much easier for a reader that is unfamiliar with the software to understand how he or she can utilize the significant efforts put into MASSyPup64.

Experimental design

No Comments

Validity of the findings

No Comments

Additional comments

Abstract is too long and ought to be shortened?

Introduction: Please remove the quotes and capital first letters for Omics, Genomics, Proteomics, etc.

The text ought to be inspected in detail and minor errors corrected. For example:

Abstract: Remove white space in the link: "http://www. bioprocess.org/massypup/" -> "http://www.bioprocess.org/massypup/".

Introduction, line 30, typo: "mass spectrometry experiments follow the all same logic" > "mass spectrometry experiments all follow the same logic"

Figure 1: I don't understand why R has been given such a big focus in the figure? After all this is just one option for the data analysis? And why is the Model Building cogwheel not connected to the others?

Raw Data Import, line 40, typo: "it is recommendable to execute" > "it is recommend to execute"

Raw Data Import, line 49, typo: "it is advisable to convert profile spectra" > "it is advised to convert profile spectra"

Reviewer 3 ·

Basic reporting

The manuscript presents a linux platform that already includes a large number of tools for mass spectrometry data analysis.
The author provides a comprehensive overview of the available software and focuses especially on further downstream ways of data analysis such as predictive models. As there already exists a paper presenting the platform, the author goes here deeper on tools for "data mining" and proposes example workflows for further data analysis.

The manuscript offers an overview of the improved platform and presents specifically designed workflows that are useful to the community. However, it still misses relevant information and clarity. Therefore, I recommend re-submission a revised version.

Issues and comments:

* The manuscript frequently states the potential of data mining for proteomics and metabolomics analysis. I am though missing a clear definition and examples of what the author means by data mining. Data mining is a very broad notion and involves practically everything from data analysis and statistics to data interpretation.

* Is there a way for upgrading to the new releases of MASSyPup? The project seems to be quite vivid with regular changes and improvements and a simple procedure could be very useful.

* line 218: "are exceed" -> "exceed"

* lines 230-231 seem to be obsolete

* line 318: "Association analysis": It is not completely clear what the study is aimed for. I suggest introduction of the biological question at the beginning. In addition, what are the input parameters? Do they include peptide FDRs?

* Data mining would include mining publicly available data. Does the package include software for GO term analysis, network/pathway analysis? Is Cytoscape installed?

* What about post-translational modifications (PTMs)? MS data often comprises large amounts of PTM measurements and they were not discussed in the manuscript. Does MASSyPub contain softwares to calculate false localization rates of PTMs on peptides? I can recommend a recently published review of PTM analysis: http://www.ncbi.nlm.nih.gov/pubmed/26216596

* What about integration with other data sources such as transcriptomics? Data mining would include retrieving additional information from databases such as Uniprot.

* There are many more R libraries applicable to MS data than presented in the manuscript, see https://www.bioconductor.org/packages/release/BiocViews.html#___MassSpectrometry. Are they installed in the platform?

* What about tools for motif search, extraction and discovery?

* I strongly recommend providing a table with all installed tools so the user can check for already available tools.

* The platform contains TPP and OpenMS. What about ProteoSaFe?

* The author applies k-means clustering amongst other techniques. Fuzzy c-means is heavily used in proteomics studies and should be worth a try, maybe yielding better separation of the chicken groups.

* The model building section (targeted metabolomics data) presents nicely different prediction models. I don't understand why one has to call the analysis "based on data mining". What is the relation to data mining here? I would rather call the analysis "predictive data modeling".
Generally, it is not clear what the author specifically means with data mining.

Experimental design

see above

Validity of the findings

see above

---

## Round 0.2 · accepted · Accept

Dear Robert,

As your submission has been initially classified as "major revisions" your revised manuscript was sent for re-evaluation to all reviewers again. As you can see in the comments below, your revised version satisfied two of the reviewers completely and I agree with their opinion. Therefore I decided to accept the manuscript without further revision.

Reviewer 2 ·

Basic reporting

As the author has chosen to completely ignore my main concern regarding the structure of the manuscript (which is still think is a major issue), I have no further comments on the revised manuscript.

Experimental design

No Comments

Validity of the findings

No Comments

Reviewer 3 ·

Basic reporting

My comments/claims were satisfactorily answered in the revised version

Experimental design

My comments/claims were satisfactorily answered in the revised version

Validity of the findings

My comments/claims were satisfactorily answered in the revised version

---

## Author Rebuttal · Round 0.2

Dr. Robert Winkler
Profesor Investigador
CINVESTAV Unidad Irapuato
Km. 9.6 Libramiento Norte Carr. Irapuato-León
36821 Irapuato Gto., México
Robert.Winkler@ira.cinvestav.mx
Tel.: +52-462-6239-635

[Figure]
October, 2015

## Response Letter to: An evolving computational platform for biological mass spectrometry: Work-flows, statistics and Data Mining with MASSyPup64

Dear Prof. Rüdiger Ettrich and estimated Referees,

First of all I want to thank all the participants of the peer review process for the rapid, diligent and fair revision of my manuscript.

Based on your comments, I stressed out more clearly the intention of facilitating the advanced data analyses in MS based -omics studies and included a table with currently installed software.

With respect to the concern of Ref. #1 about the presumed high false-discovery rate (FDR), I agree that even theoretically unlimited mass accuracy is insufficient to unequivocally identify metabolites (Kind, T.; Fiehn, O. Metabolomic database annotations via query of elemental compositions: Mass accuracy is insufficient even at less than 1 ppm. BMC Bioinformatics 2006, 7 (1), 234.), which might lead to numerous false metabolite identifications.

But on the other side, currently - contrary to peptide/protein identifications – no algorithm is available to statistically estimate the FDR of metabolite identifications.

SpiderMass has a novel approach. First, a compound database for the specific project is built from experimental data and/or potential metabolites from genomics projects. Then, the experimental data are matched against this database, using both, mass and isotope distribution pattern information. The SpiderMass program integrates the 'Seven Golden Rules' (Kind, T.; Fiehn, O. Seven Golden Rules for heuristic filtering of molecular formulas obtained by accurate mass spectrometry. BMC Bioinformatics 2007, 8, 105.), with de-novo formula generartor, as well as a database generator/ matching algorithm and an online-search (Winkler, R. SpiderMass: Semantic database creation and tripartite metabolite identification strategy. Journal of Mass Spectrometry 2015, 50 (3), 538–541.).

The SpiderMass software was already employed successfully in various published metabolomics studies (Palmeros-Suárez, P. A.; Massange-Sánchez, J. A.; Martínez-Gallardo, N. A.; Montero-Vargas, J. M.; Gómez-Leyva, J. F.; Délano-Frier, J. P. The overexpression of an Amaranthus hypochondriacus NF-YC gene modifies growth and confers water deficit stress resistance in Arabidopsis. Plant Science 2015, 240, 25–40.; Gamboa-Becerra, R.; Ramírez-Chávez, E.; Molina-Torres, J.; Winkler, R. MSI.R scripts reveal volatile and semi-volatile features in low-temperature plasma mass spectrometry imaging (LTP-MSI) of chilli (Capsicum annuum). Anal Bioanal Chem 2015, 407 (19), 5673–5684.; Sotelo-Silveira, M.; Chauvin, A.-L.; Marsch-Martínez, N.; Winkler, R.; De Folter, S. Metabolic fingerprinting of

Arabidopsis thaliana accessions. Frontiers in Plant Science 2015, 6 (365).), which underlines the practical usability of the identifications.

If, after reading the comments and corrections, the reviewers still suggest improving the clarity, I would suggest to the editor the use of numbered sections to facilitate the understanding the structure of the manuscript, which contains: 1) General concepts of MS data analysis, Statistics and Data Mining and workflow-design, 2) Design of a computational platform and 3) Practical examples.

Following I will respond to the further comments of the reviewers point-by-point:

## Reviewer Comments

## Reviewer 1 (Ben Bowen)

**BB**: Getting a large number of packages to work together on a single operating system is a daunting task. Configuring specific compilers, interpreters, and organization of permissions is often not achievable by typical scientists. The evolution of MassyPup64 is an important development to encourage distribution of tools that can seemlessly work together.

**RW**: Thank you very much for your positive opinion about my intend to make MS data processing tools more accesible to typical users.

**BB**: Recommended revisions:
Original data is almost always in a proprietary format from a vendor. Elaborate on line 44 which formats are compatible with msconvert via the MassyPup64 Linux distribution and which are not. Its my naieve understanding that certain vendors have released their dlls in such a way that they are compatible with Wine and other Linux .net emulators. This is an excellent opportunity to identify the current state of affairs for raw data import for each vendor.

**RW**: Technically speaking, some vendor libraries might be usable with emulators. But the major MS companies do not allow the distribution of their software. I think, this classic and annoying problem could be discussed an opinion article. But from a practical point of view, the raw data should be converted to .mzML directly on the ms control computer, as recommended in the manuscript. Further conversions are no problem with msconvert.

**BB**: imzml/ibd files are mentioned in figure 1, but there is no mention of mass spectrometry imaging other than line 234 "MSI.R for evaluating MSI data." There are numerous software packages for processing imzml, and they aren't mentioned. This should be corrected unless there is a reason that they can't be installed on MassyPup64?

**RW**: I tried various programs for MSI data analysis and after some frustration I wrote the MSI.R scripts, which are mentioned and installed on MASSyPup64. Since the employment of R has many advantages (parallel computing, plenty of graphics and statistics packages, libraries for MS data processing etc.), I did not install other programs, which often have critical limitations (DataCube Exporer is a pre-compiled .exe, the imzML Converter is not really open source, others require commercial MATLAB or LabView libraries, Cardinal only processes profile data). Therefore I did not install and mention them, although they might be very useful on different platforms/ environments.

**BB**: Python is mentioned briefly on line 205 and there is absolutely no mention of the IPython interface. Including the Python packages for mass spectrometry, computational chemistry, and scientific computing packages needed for plotting, statistics, and file import/export seems like a major oversight that could be easily overcome.

**RW**: Indeed, I was not aware of IPython, but it looks very useful for advanced scientific programming. Is there any example about its employment in MS data analysis? With pleasure I can integrate it in MASSyPup64, if requested. For our lab projects, the workflow system taverna seems to fulfil the necessary functions.

**BB**: It isn't clear from reading the manuscript if an MSMS reference database (ie: from mass bank or Metlin) and a search algorithm for comparing a measured MSMS spectrum to a reference database is included.

**RW**: Currently, no MS/MS database is included. METLIN seems not to be available for public download. At the moment, we are testing the local installation of a MassBank server, since the online service had problems in the last months. If everything works fine, I'll try an installation of a MassBank search on MASSyPup64.

**BB**: It isn't clear if mzmine is included in this release. For example, mzmine is mentioned on line 162. Is it included? Is Java included? Is the java version compatible with other java apps in MassyPupp64?
My general impression is that many valuable tools are bundled together in this linux distribution, but the style and organization of the paper in its current form makes it very difficult to tell exactly what is included and what it can be used for. A full page table listing the software included would make this much clearer.

**RW**: MZmine and Java are included in MASSyPup64. The revised manuscript contains a table with available software on MASSyPup64 and their uses.

**BB**: I was surprised to see the mention of putative-identification on line 503-505. I did not see any mention of the consolidation of features into pure-spectra where adducts, isotopes, and degradation products can be identified. It is a concern that this this manuscript advocates for accurate mass based assignments of features to compounds independent of pure-spectra and MSMS. This may not be the viewpoint of the author, but concluding the paper on this point makes it seem like a highlight. Addressing the extremely high false-discovery rate in this approach is necessary, at a minimum.

**RW**: The identification strategy of SpiderMass is pretty novel, since it combines heuristic tools (based on the Seven Golden Rules by Kind & Fiehn), online-searches and – most importantly – biological knowledge (database generator, which is fed with already identified compounds or theoretical metabolites from genomics projects). Since I already published an article about the program recently (Winkler, 2015), I would not like to extend the part about the program again. I hope, you are fine with this. But focussing on relevant compounds by using a context-specific database should drastically reduce false identifications.

## Reviewer 2 (Anonymous)

**R2**: The manuscript by Robert Winkler details a computational platform for mass spectrometry analysis called MASSyPup64, which is an evolution on the already published MASSyPup platform. But while the work on providing easy access to open source bioinformatics software is highly commendable, the manuscript in its current form leaves me rather confused.

To me it seems like the author is trying to write multiple manuscripts in one, and I think that readers not familiar with most of the software used will be unable to extract much from the text. I'd recommend restructuring and refocusing the manuscript into more of a technical brief kind of manuscript, with a lot of the current details and examples moved to either a supplementary or the MASSyPup64 web page. I think this will make it much easier for a reader that is unfamiliar with the software to understand how he or she can utilize the significant efforts put into MASSyPup64.

**RW**: During the structuring of the manuscript I also was thinking a lot about the adequate presentation of the content. There are three main components: 1) Introduction of the concepts "General data processing in mass spectrometry", "Workflow design", "Statistics and Data Mining" etc. 2) Implementation of a computational platform to perform such analyses, 3) Presentation of practical examples, which demonstrate the benefit by using workflow and Data Mining. I appreciate that reading the text the first time (without background in mass spectrometry data analysis) could be somehow confusing, but shortening the content would reduce the informative value for the reader. Actually, my main motivations to send the manuscript to PeerJ were the possibility to publish a large article as well as to make it available publicly. If the editor agrees, I would suggest to structure the content with numbered sections.

**R2**: Abstract is too long and ought to be shortened?

**RW**: The abstract complies with the specifications of PeerJ.

**R2**: Introduction: Please remove the quotes and capital first letters for Omics, Genomics, Proteomics, etc.

**RW**: Genomics, transcriptomics, metabolomics and proteomics are now in sentence style throughout the manuscript. Only the quotes of the first mention of '-omics' was left.

**R2**: The text ought to be inspected in detail and minor errors corrected. For example: Abstract: Remove white space in the link: "http://www. bioprocess.org/massypup/" -> "http://www.bioprocess.org/massypup/".

**RW**: Sorry, I did not find the white space. The link works fine on my computer.

**R2**: Introduction, line 30, typo: "mass spectrometry experiments follow the all same logic" > "mass spectrometry experiments all follow the same logic"

**RW**: Thanks; corrected.

**R2**: Figure 1: I don't understand why R has been given such a big focus in the figure? After all this is just one option for the data analysis? And why is the Model Building cogwheel not connected to the others?

**RW**: The figure is analogous to a calender arrangement. R is very important in the process of model building. The figure should express that the first four steps are pretty mechanic. In contrast, the last step, creating meaningful (biological) models from the data using R is, what everything is about. Other programs could fulfil the same purpose, but R is very popular among quantitative biologists, mature, provides plenty of packages and publicly available.

**R2**: Raw Data Import, line 40, typo: "it is recommendable to execute" > "it is recommend to execute"

**RW**: Thanks, corrected.

**R2**: Raw Data Import, line 49, typo: "it is advisable to convert profile spectra" > "it is advised to convert profile spectra".

**RW**: Thanks, corrected.

## Reviewer 3 (Anonymous)

**R3**: The manuscript presents a linux platform that already includes a large number of tools for mass spectrometry data analysis.
The author provides a comprehensive overview of the available software and focuses

especially on further downstream ways of data analysis such as predictive models. As there already exists a paper presenting the platform, the author goes here deeper on tools for "data mining" and proposes example workflows for further data analysis.

The manuscript offers an overview of the improved platform and presents specifically designed workflows that are useful to the community. However, it still misses relevant information and clarity. Therefore, I recommend re-submission a revised version.

**RW**: Dear Reviewer 3, thanks for your comments. I will try to resolve your doubts.

**R3**: The manuscript frequently states the potential of data mining for proteomics and metabolomics analysis. I am though missing a clear definition and examples of what the author means by data mining. Data mining is a very broad notion and involves practically everything from data analysis and statistics to data interpretation.

**RW**: The definition of Data Mining is covered in the introduction: "Data Mining enhances 'classic' Statistics methods with machine learning ('artificial intelligence') algorithms and computer science. Data Mining supports the understanding of complex systems, which contain wealth of data with interacting variables. An important aspect of DM is the development of models, which represent the data in a structured form and support the extraction of information and creation of knowledge (Williams, 1987, 1988, 2011)" etc. Please indicate, if any more explanation is required.

**R3**: Is there a way for upgrading to the new releases of MASSyPup? The project seems to be quite vivid with regular changes and improvements and a simple procedure could be very useful.

**RW**: Each release is a complete compilation consisting of operating system and programs. Therefore, it is not necessary to upgrade individual files. The user settings, programs and data can be stored locally and are kept on future releases. This mechanism is now explained in the manuscript.

**R3**: line 218: "are exceed" -> "exceed"

**RW:** Thanks, corrected.

**R3**: lines 230-231 seem to be obsolete

**RW**: Indeed, lines 230-231 were removed.

**R3**: line 318: "Association analysis": It is not completely clear what the study is aimed for. I suggest introduction of the biological question at the beginning. In addition, what are the input parameters? Do they include peptide FDRs?

**RW**: At the end of the Bioanalytical Question we want to "suggest peptides for a targeted screening of peroxidases". This is not trivial, since the identification of peptides depends on many technical and non-technical parameters. The input data and parameters are given in the manuscript below "Association Analysis", peptides with a probability of at least 0.9 were considered.

**R3**: Data mining would include mining publicly available data. Does the package include software for GO term analysis, network/pathway analysis? Is Cytoscape installed?

**RW**: DM does not necessarily include public data. Unfortunately, MS experiments from different labs or experiments are often difficult to compare. Cytoscape nor GO term analysis programs are currently installed, but any software that is compatible with Java, Python, C/C++, Perl or other open standards can be installed without problems and/or integrated with taverna.

**R3**: What about post-translational modifications (PTMs)? MS data often comprises large amounts of PTM measurements and they were not discussed in the manuscript. Does MASSyPub contain softwares to calculate false localization rates of PTMs on peptides? I can recommend a recently published review of PTM analysis: http://www.ncbi.nlm.nih.gov/pubmed/26216596

**RW:** The bundled peptide/ protein search engines (please see table 1 in the revised manuscript) all enable the search for PTMs. The validity of potential PTM can be investigated with PeptideProphet/ ProteinProphet, PTM explorer etc.

**R3**: What about integration with other data sources such as transcriptomics? Data mining would include retrieving additional information from databases such as Uniprot.

**RW**: This could be done using a taverna workflow.

**R3**: There are many more R libraries applicable to MS data than presented in the manuscript, see https://www.bioconductor.org/packages/release/BiocViews.html#___MassSpectrometry. Are they installed in the platform?

**RW**: Yes. The R packages installed (and listed now in table 1) depend on plenty of libraries for MS analysis. XCMS/metaXCMS and MSI.R permit the programming of complete workflows (from raw data processing to statistical evaluation, graphics and metabolite identification).

**R3**: What about tools for motif search, extraction and discovery?

**RW**: Those could be integrated into a taverna workflow from different online resources without problem.

**R3**: I strongly recommend providing a table with all installed tools so the user can check for already available tools.

**RW**: The revised manuscript contains a table with available software on MASSyPup64 and their function.

**R3**: The platform contains TPP and OpenMS. What about ProteoSaFe?

**RW**: ProteoSAFe is available as an online resource. Therefore, the installation of another proteomics workflow system would be redundant.

**R3**: The author applies k-means clustering amongst other techniques. Fuzzy c-means is heavily used in proteomics studies and should be worth a try, maybe yielding better separation of the chicken groups.
The model building section (targeted metabolomics data) presents nicely different prediction models. I don't understand why one has to call the analysis "based on data mining". What is the relation to data mining here? I would rather call the analysis "predictive data modeling".
Generally, it is not clear what the author specifically means with data mining.

**RW**: The creation of predictive models and the extraction of non-obvious correlations are important aspects of Data Mining. This is explained extensively in the introduction. Both, the chicken and the Arabidopsis experiments, demonstrate that applying e.g. the Random Forest model builder leads to reliable models for classfication with little effort by the user.

I hope that I answered all questions to your satisfaction, and that the manuscript now is suitable for publication in *PeerJ*.

In case of any further doubts or questions, please do not hesitate to contact me.

Yours sincerely,

Prof. Dr. Robert Winkler